# Anti-ApoA-1 IgGs in Familial Hypercholesterolemia Display Paradoxical Associations with Lipid Profile and Promote Foam Cell Formation

**DOI:** 10.3390/jcm8122035

**Published:** 2019-11-21

**Authors:** Sabrina Pagano, Alessandra Magenta, Marco D’Agostino, Francesco Martino, Francesco Barillà, Nathalie Satta, Miguel A. Frias, Annalisa Ronca, François Mach, Baris Gencer, Elda Favari, Nicolas Vuilleumier

**Affiliations:** 1Division of Laboratory Medicine, Department of Genetics and Laboratory Medicine, Geneva University Hospital, 4 rue Gabrielle-Perret-Gentil, 1205 Geneva, Switzerland; Nathalie.Satta@unige.ch (N.S.); Miguel.Frias@unige.ch (M.A.F.); nicolas.vuilleumier@hcuge.ch (N.V.); 2Department of Internal Medicine Specialities, Medical Faculty, Geneva University, 1 rue Michel Servet, 1206 Geneva, Switzerland; 3Fondazione Luigi Maria Monti, Istituto Dermopatico dell’ Immacolata-IRCCS, Experimental Immunology Laboratory, Via dei Monti di Creta 104, 00167 Rome, Italy; ale.magenta@gmail.com (A.M.); marcodagostino86@hotmail.it (M.D.); 4Department of Pediatrics, Sapienza University of Rome, Viale Regina Elena 324, 00161 Rome, Italy; Francesco.Martino@uniroma1.it; 5Department of Cardiovascular, Respiratory, Nephrological, Anesthesiological and Geriatrical Sciences, Sapienza University of Rome, Viale del Policlinico 155, 00161 Rome, Italy; Francesco.Barilla@uniroma1.it; 6Department of Food and Drug, University of Parma, Parco Area delle Scienze 27/A, 43124 Parma, Italy; annalisa.ronca@studenti.unipr.it (A.R.); elda.favari@unipr.it (E.F.); 7Division of Cardiology, Geneva University Hospital, 1205 Geneva, Switzerland; Francois.Mach@hcuge.ch (F.M.); Baris.Gencer@hcuge.ch (B.G.)

**Keywords:** anti-apolipoprotein A-1 IgG, familial hypercholesterolemia, cholesterol homeostasis, foam cells, miR-33a, TLR2/4, passive diffusion

## Abstract

Aims: Anti-Apolipoprotein A-1 autoantibodies (anti-ApoA-1 IgG) promote atherogenesis via innate immune receptors, and may impair cellular cholesterol homeostasis (CH). We explored the presence of anti-ApoA-1 IgG in children (5–15 years old) with or without familial hypercholesterolemia (FH), analyzing their association with lipid profiles, and studied their in vitro effects on foam cell formation, gene regulation, and their functional impact on cholesterol passive diffusion (PD). Methods: Anti-ApoA-1 IgG and lipid profiles were measured on 29 FH and 25 healthy children. The impact of anti-ApoA-1 IgG on key CH regulators (SREBP2, HMGCR, LDL-R, ABCA1, and miR-33a) and foam cell formation detected by Oil Red O staining were assessed using human monocyte-derived macrophages. PD experiments were performed using a validated THP-1 macrophage model. Results: Prevalence of high anti-ApoA-1 IgG levels (seropositivity) was about 38% in both study groups. FH children seropositive for anti-ApoA-1 IgG had significant lower total cholesterol LDL and miR-33a levels than those who were seronegative. On macrophages, anti-ApoA-1 IgG induced foam cell formation in a toll-like receptor (TLR) 2/4-dependent manner, accompanied by NF-kB- and AP1-dependent increases of SREBP-2, LDL-R, and HMGCR. Despite increased ABCA1 and decreased mature miR-33a expression, the increased ACAT activity decreased membrane free cholesterol, functionally culminating to PD inhibition. Conclusions: Anti-ApoA-1 IgG seropositivity is frequent in children, unrelated to FH, and paradoxically associated with a favorable lipid profile. In vitro, anti-ApoA-1 IgG induced foam cell formation through a complex interplay between innate immune receptors and key cholesterol homeostasis regulators, functionally impairing the PD cholesterol efflux capacity of macrophages.

## 1. Introduction

Humoral autoimmunity has recently been shown to represent a dual mediator of atherogenesis and cardiovascular diseases (CVD) by modulating three main pathways, including inflammation, coagulation, and foam cell formation [1,2]. Among autoantibodies of interest in CVD, the interest in antibodies against apolipoprotein A-1 (anti-ApoA-1 IgG) appears to be gaining momentum. Indeed, three recent studies derived from a large multicenter general population cohort demonstrated that anti-ApoA-1 IgGs were an independent cardiovascular (CV) risk factor predictive of poor prognosis [3,4,5] similarly to what had been reported previously and more recently in high CV risk populations [6,7,8,9,10,11]. In parallel, translational studies pointed to these autoantibodies as mediators of atherogenesis, promoting atherosclerosis, myocardial necrosis, and mice death through toll-like receptors (TLR) 2,4 and CD14 signaling [12,13,14]. Interestingly, inverse associations were regularly noted between anti-ApoA-1 IgG, total cholesterol, high-density lipoprotein (HDL), and low-density lipoprotein (LDL) levels [3,13,15,16], potentially suggesting that these antibodies could also interfere with cholesterol metabolism in addition to their established pro-inflammatory and pro-thrombotic properties [13,14,17]. Such a hypothesis has been further supported by two recent observations. The first showed that antibodies directed against the c-terminal part of ApoA-1 in type 2 diabetes were associated with decreased cholesterol efflux capacity (CEC) of fibroblasts [18]. The second reported that anti-ApoA-1 IgG levels were inversely associated with PD- and positively associated with ABCA1-CEC in healthy obese subjects, with the capacity, to enhance ABCA1-CEC in vitro while repressing PD-CEC, leading to foam cell formation [19], the hallmark of atherosclerosis [20]. Nevertheless, the exact molecular mechanisms underlying these drastic cellular phenotype modifications induced by anti-apoA-1 IgG are still elusive, and likely to imply major changes in the gene and protein expression levels of key regulators of cellular cholesterol homeostasis; whether such effect could be reproduced in human macrophages is currently unknown [20,21,22,23].

Among these, 3-hydroxy-3-methylglutaryl CoA reductase (HMGCR) and low-density lipoprotein receptor (LDL-R) are known as key cholesterol homeostasis regulatory elements leading to the developments of statins, the most efficient and commonly used pharmacological molecules available to prevent atherosclerosis-related complications [21]. HMGCR and LDLR are controlled by the sterol regulatory element binding protein 2 (SREBP2) encoded by *SREBF2* gene, which contains microRNAs 33a (miR-33a) in its intronic sequence and reduces cholesterol efflux via repression of the adenosine triphosphate (ATP)-binding cassette transporter A1 (ABCA1), impairing HDL biogenesis [22,23]. Upon *SREBF2* gene activation, both LDLR and HMGCR are upregulated by SREBP2, enhancing LDL uptake and increasing intracellular cholesterol synthesis. In response to *SREBF2* gene transcription, miR-33a expression will increase, inhibiting ABCA-1 cholesterol efflux [22,23]. Therefore, miR-33a and SREBP2 act synergistically to efficiently increase cellular cholesterol levels by inhibiting cellular cholesterol efflux and increasing lipid uptake and intracellular synthesis, respectively. While previous studies suggested that miR-33 may represent therapeutic target for the treatment of cardiovascular disease [24,25], a recent work showed that miR-33 deletion in mice results in dyslipidemia, obesity, and insulin resistance [26], assigning a role for miR-33 that is much more complex than what has been considered so far. Such complexity has also been illustrated in humans by the fact that miR-33 levels are surprisingly elevated in pediatric patients affected by familial hypercholesterolemia (FH) [27].

Since foam cells can be generated by the uncontrolled uptake of unmodified LDL via LDL-R [28,29], any factor influencing LDLR, HMGCR, or miR-33a expression will efficiently modulate lipid uptake, cellular cholesterol synthesis, and cholesterol efflux, and therefore affect atherogenesis.

Taken together, these observations suggest that anti-ApoA-1 IgG may reorient the lipids from the plasmatic compartment toward the intracellular lipid pools, potentially explaining the paradoxical associations frequently retrieved between anti-apoA-1 IgG and lipid profile despite an increased CV risk. If so, such a hypothesis would imply modulation of key regulators of cellular cholesterol homeostasis and foam cell formation, including miR-33.

Therefore, in this translational study, we explored the associations between anti-ApoA-1 IgG, lipid profile, and miR-33a levels in FH children as an extreme and optimal human dyslipidemia phenotype to perform such explorations and compared whether such associations could be retrieved in age- and gender-matched controls. Fueled by these clinical observations, we then dissected the in vitro mechanisms by which anti-ApoA-1 IgG could lead to foam cell formation. Our results showed that associations between anti-ApoA-1 IgG, lipid profile, and miR-33 levels were only retrieved in FH children. Furthermore, our in vitro studies using human macrophages provided the first molecular insights on how these antibodies could act as novel endogenous cholesterol homeostasis disruptors.

## 2. Experimental Section

### 2.1. Patients

In this study, we considered 29 children (14 males and 15 females) aged 5–15 affected by FH and referred to the Center of Clinical Lipid Research, Department of Pediatrics, Sapienza University of Rome [27]. Children were classified as FH on the basis of the presence of a first-degree relative with hypercholesterolemia (total cholesterol (TC) >95th age and sex-specific percentile) according to the MEDPED criteria [30]. In the same center, a control group of 25 healthy children (11 males, 14 female) was recruited with a BMI (body mass index) appropriate for their age and gender, as matched for age with the FH group.

Exclusion criteria consisted of acute or chronic disease or infection (connective tissue disease, hypothyroidism, renal disease, malignancy, clinical evidence of CVD, diabetes mellitus, hypertension or metabolic syndrome), autoimmune disease, or the use of medication potentially affecting growth and development, associated with a history of alcohol consumption and smoking (when appropriate), immunosuppressive drugs, non-steroidal, anti-inflammatory, lipid-lowering drugs, and/or vitamin supplements.

At first visit, anthropometric data were measured (body weight and waist, hip, and arm circumferences). Weight was measured using an electronic scale (Soehnle, Murrhardt, Germany) and standing height was measured with the Harpenden Stadiometer (Holtain, Crymych, UK). Systolic and diastolic blood pressure was measured using a random zero sphygmomanometer (Hawksley & Sons Ltd., Lancing, UK); the mean of three measurements was used in the analysis. BMI was calculated as weight/height^2^ (kg/m^2^).

Informed written consent was obtained from all the participants. The study was in conformity with the ethical guidelines of the Declaration of Helsinki, and was reviewed and cleared by the Ethical Committee of Sapienza University of Rome.

### 2.2. Plasma Samples and Blood Analyses

Venous blood samples (10 mL) were collected in EDTA-containing tubes from 12 h fasted subjects. Blood was then centrifuged (1200*g* for 10 min at 4 °C). Supernatant was then collected and centrifuged (2000*g* for 10 min at 4 °C). Plasma samples were stored at −80 °C and were thawed on ice before use. Plasma concentrations of lipoprotein and apolipoproteins were determined as previously described [27]. LDL cholesterol was calculated using Friedewald’s equation.

### 2.3. Assessment of Anti-ApoA-1 IgG Levels

Anti-ApoA-1 IgG were measured as previously described [7,9,13,14,17]. Briefly, MaxiSorp plates (Nunc^TM^, city, Roskilde, Denmark) were coated with purified, human-derived delipidated apolipoprotein A-1 (20 μg/mL; 50 μL/well) for 1 h at 37 °C. After being washed, all wells were blocked for 1 h with 2% bovine serum albumin (BSA) in a phosphate-buffered solution (PBS) at 37 °C. FH samples were also added to a non-coated well in order to assess individual non-specific binding. After six washing cycles, 50 μL/well of the alkaline phosphatase-conjugated anti-human IgG was added (Sigma-Aldrich, St Louis, MO, USA), it was diluted at 1:1000 in a PBS/BSA 2% solution, and this was added and incubated for 1 h at 37 °C. After washing six more times, phosphatase substrate p-nitrophanylphosphate disodium (Sigma-Aldrich) dissolved in a diethanolamine buffer (pH 9.8) was added and incubated for 30 min at 37 °C. Optical density (OD) was determined at 405 nm (Filtermax 3, Molecular Devices^TM^, San Jose, CA, USA). and each sample was tested in duplicate. Corresponding non-specific binding was subtracted from mean OD for each sample. The specificity of detection was assessed using conventional saturation tests by Western blot analysis.

As previously described, elevated levels of anti-ApoA-1 IgG (seropositivity) were defined by an OD cutoff of OD > 0.64 corresponding to the 97.5th percentile of a reference population. In order to limit the impact of interassay variation, we further calculated an index consisting of the ratio between sample net absorbance and the positive control net absorbance × 100. The index value corresponding to the 97.5th percentile of the normal distribution was 37. Accordingly, to be considered as positive (presenting elevated anti-apoA-1 IgG levels), samples had to display both an absorbance value of OD > 0.64 and an index value ≥37%. [7,9,13,14,17].

### 2.4. Reagents

RPMI-1640 medium, fetal bovine serum (FBS), PBS free of Ca^2+^ and Mg^2+^, L-glutamine, penicillin, and streptomycin were obtained from Gibco BRL-Life Technologies (Rockville, MD, USA). Interferon-gamma (IFN-γ) was from Roche (Mannheim, Germany).

Affinity purified goat polyclonal anti-human ApoA-1 IgG (ref. 11AG2) was obtained from Academy Bio-Medical Company (Houston, TX, USA) and goat control IgG were from Meridian Life Science (ref. A66200H) (Saco, ME, USA). Ultrapure lipopolysaccharide (LPS) from *Escherichia coli* was purchased from Alexis Enzo Life Sciences (Lausen, Switzerland). Blocking anti-human TLR4 (clone HTA 125) antibody, anti-human TLR2 antibody (clone TL2.1), and matched isotype control antibodies were from Biolegend (San Diego, CA, USA). Blocking anti-human TLR2 (clone TL2.5) and blocking anti-human CD14 antibodies were from InvivoGen (San Diego, CA, USA). SP600125 (c-Jun N-terminal kinase (JNK) inhibitor and BAY11-7082 (IkB-α inhibitor) were from InvivoGen. San Diego, CA, USA)

### 2.5. Human Monocyte-Derived Macrophage Preparation

Human monocytes were isolated from buffy coats obtained from healthy donors in the Geneva Hospital Blood Transfusion Center (Geneva, Switzerland) and differentiated into macrophages by 24 hours incubation with IFN-γ (500 U/mL) in a complete RPMI-1640 culture medium (10% heat-inactivated FCS, 50 µg/mL streptomycin, 50 U/mL penicillin, 2 mM L-glutamine) as previously described [14,17]. Macrophage preparation consisted of >90% CD68+ cells as assessed by flow cytometry.

When indicated, anti-TLR4, anti-TLR2 (TL2.5/2.1), and anti-CD14 blocking antibodies, as well as isotype-matched control mAb (10 µg/mL) and specific pharmacological inhibitors, were added 30 min before stimulation with IgG [14,17].

### 2.6. Protein Purification and Western Blot Analysis

Macrophages (2 × 10^6^) were lysed at 4 °C for 20 min with radioimmunoprecipitation assay (RIPA) lysis buffer with added protease inhibitors (Complete tablets, mini, Roche diagnostics, Mannheim, Germany) and phosphatase inhibitors (Halt Protease and Phosphatase Inhibitor Cocktail, Thermo Scientific, Waltham, MA, USA) and then samples were centrifuged at 14,000 rpm at 4 °C for 15 min. Protein concentration was determined by Bradford protein assay (Biorad, Hercules, CA, USA). To increase protein concentration, samples were loaded onto Amicon Ultra-0.5 centrifugal filter devices 3K (Merck Millipore, Darmstadt, Germany) following the manufacturer’s instructions. Forty micrograms of total protein extract were resolved by 8% or 10% polyacrylamide gel electrophoresis under reducing conditions, and transferred to a polyvinylidene difluoride (PVDF) membrane (Immobilon, Millipore IPVH 00010). Membranes were incubated with the following antibodies: anti-SREBP2 (rabbit polyclonal, ref. 10007663, Cayman chemical, Ann Arbor, MI, Stati Uniti), anti-LDL Receptor (rabbit monoclonal, ref. ab52818, Abcam, Cambridge, UK), anti-HMGCR (rabbit monoclonal, ref. ab174830, Abcam), anti-SCAP (Rabbit polyclonal, ref. ab125186 Abcam), anti-ABCA1 (rabbit polyclonal, ref. NB400-105, Novus Biologicals, Centennial, CO, USA), anti-β-actin (mouse monoclonal, ref. ab8226, Abcam). Horseradish peroxidase-conjugated antisera from Dako (Glostrup, Denmark), were used to reveal primary antibody binding, with detection by BM Chemiluminescence Blotting Substrate (POD) from Roche Diagnostics (Mannheim, Germany). Relative protein levels were measured using densitometric analysis with ImageJ software (1.48v, Java 1.5.0_20, 64-bit). Results are expressed in arbitrary units.

### 2.7. Lipid Uptake by Oil Red O Staining

Monocytes (3 × 10^5^) were plated onto Lab-Tek chamber slide system (Nunc, Roskilde, Denmark) and IFN-γ (500 U/mL) was added to RPMI supplemented with 10% FCS for 24 h at 37 °C. After that time, cells were washed once with PBS and exposed, or not, for 24 h to 20 μg/mL LDL from Academy Bio-Medical Company (Houston, TX, USA). One hour before adding LDL, cells were incubated with 40 μg/mL of goat polyclonal anti-human apolipoprotein A-1 or control IgG, shown to be the optimal dose to elicit a pro-inflammatory response [7,9,13,14,17]. Cells were examined for lipid inclusion by Oil Red O staining. Briefly, cells were incubated with 10% formalin (Sigma-Aldrich, St Louis, MO, USA) for 30 min at room temperature, incubated with Oil Red O solution (Sigma- Aldrich, St Louis, MO, USA) for 15 min, and cells were then counterstained with hematoxylin (Sigma- Aldrich, St Louis, MO, USA). Aquatex (Merck Millipore, Darmstadt, Germany) was used as mounting media. Images were acquired with a microscope Zeiss Axioskop 2 plus. We used a 40× objective (Plan-Neofluar 40× /0.75 Ph2) for all images. The images were collected using the AxioVision 4.8.1.0 softwares (Zeiss, Oberkochen, Germania). Quantification of the lipid content (Oil Red O staining) per cell, identified by hematoxylin blue nuclear staining, was performed using Definiens Developer XD2 (Cambridge, MA, USA). Results are expressed as mean granule area in arbitrary units.

### 2.8. RNA Extraction

Macrophage cells (1 × 10^6^) were subjected to RNA extraction using a total RNA purification RNeasy Micro kit (Qiagen, Hilden, Germany) according to the manufacturer’s protocol. RNA from plasma (200 μL) was extracted using a Total RNA Purification Plus kit (Norgen Biotek, Thorold, ON, Canada) and RNA from macrophage supernatant (200 μL) was extracted using TRIzol LS (Ambion, Life-Technologies, Austin, TX, USA) as an internal control, with 10 fmol cel-miR-39a (Qiagen, Hilden, Germany) spiked into each plasma and supernatant sample after adding lysis buffer or TRIzol LS. We then followed the manufacturer’s protocol for RNA extraction. RNA was quantified using NanoDrop 2000 software (NanoDrop Products, Thermo Fisher Scientific, Wilmington, DE, USA.).

### 2.9. miRNA Quantitative PCR

miRNA quantification was done using a TaqMan^®^MicroRNA Reverse Trancription Kit (Applied Biosystems) for reverse transcription reactions, according to the manufacturer’s instructions. TaqMan Universal PCR master mix (Life Technologies, Carlsbad, CA, USA) was used to quantify miRNA levels in plasma, macrophage cells, and in the supernatant counterpart with a 7900HT SDS Fast System instrument (Applied Biosystems, Foster City, CA, USA). Primers for *miR-33a, cel-miR-39a* were obtained from Applied Biosystems (Foster City, CA, USA). *miR-33a* level in plasma and in cell supernatant were normalized to the spiked *cel-miR-39a* whereas *miR-33a* expression levels in each sample from macrophage cells were normalized to RNU6B, miR-16, Z30 controls (Applied Biosystems, Foster City, CA, USA). geNorm was used to determine the most stable genes from the set of control tested genes since geNorm calculates the gene expression stability measure (M value) for a control gene as the average pairwise variation for that gene with all other tested control genes [31]. Data were analyzed with the 7900HT SDS Software v2.3 (Applied Biosystems, Foster City, CA, USA). Relative expressions of *miR-33a* were calculated using the comparative threshold cycle values (*C*_T_) method (2^−ΔΔ^*C*t) [32].

### 2.10. Quantitative PCR

For mRNA quantification, cDNA was synthesized using the High Capacity RNA to cDNA kit (Life Technologies, Carlsbad, CA, USA). Quantitative re al-time PCR was performed in duplicate using TaqMan Universal Master mix II, no UNG on the StepOne Plus Real-Time PCR System (Thermo Fisher Scientific, Waltham, MA, USA). The mRNA levels were normalized to Gapdh (hs99999905_m1) as a housekeeping gene. The following primers were used: pri-miR-33a (hs03293451), Abca1 (hs01059137_m1), and Srebf-2 (hs01081784_m1). All primers were obtained from (Applied Biosystems, Foster City, CA, USA). Data were analyzed with the 7900HT SDS Software v2.3 (Applied Biosystems, Foster City, CA, USA). Relative expression of mRNA was calculated using the comparative threshold cycles values (*C*_T_) method (2^−ΔΔ^*C*t) [32].

### 2.11. Anti-ApoA-1 IgG Modulation of Membrane Free Cholesterol. Assay of Cholesterol Oxidase

Cholesterol oxidase treatment was essentially as previously described [33]. Briefly, cells were labeled with 3 μCi/mL [3 H] cholesterol. Cholesterol oxidase (1 U/mL) was added, and cells were incubated for 4 hours. Lipid was extracted with isopropanol, and radioactive cholesterol and cholestenone were separated using thin-layer chromatography and quantified.

### 2.12. ACAT Activity Assessment

Cholesterol esterification was evaluated as the incorporation of radioactivity into cellular cholesteryl esters after addition of [14C]-oleate-albumin complex [34]. At the end of incubation, cells were washed with PBS and lipids were extracted with hexane/isopropanol (3:2). The extracted lipids were separated by TLC (isooctane/diethyl ether/acetic acid, 75:25:2 (v/v/v)). Cholesterol radioactivity in the spots was determined by liquid scintillation counting.

### 2.13. Measurement of Free Cholesterol Content in Cell Supernatant

Free cholesterol in supernatant was measured using the fluorometric method using the Cholesterol Quantitation Assay Kit by Abcam (ab65359, Cambridge, UK) following the manufacturer’s instruction. The experiments were conducted without ACAT inhibitors.

### 2.14. Passive Diffusion Analysis

We used a validated model of THP-1 monocytes cultured in RPMI 1640 medium supplemented with 10% FBS at 37 °C in 5% CO_2_ [35]. To perform the experiments for passive diffusion, cells were seeded in 24-well plates at a density of 5 × 10^5^ cells/well in the presence of 100 ng/mL PMA for 72 h to allow differentiation into macrophages. Cells were labeled with (1,2-3H)-cholesterol in the presence of an ACAT inhibitor (2 µg/mL, Sandoz 58035 by Sandoz, Holzkirchen, Germany) for 24 h. In order to measure passive diffusion, the key cholesterol efflux regulator in non-dyslipidemic macrophages [36], the experiment was performed in the absence of 10% whole serum, but cells were supplemented with 0.2% albumin to avoid any active ABCA1- or ABCG1-mediated cholesterol efflux. Cholesterol measurement was expressed as a percentage of the radioactivity released to the medium in 4 h over the total radioactivity incorporated by cells. Control samples were run to confirm the responsiveness of cells. Background efflux, evaluated in the absence of acceptors, was subtracted from each samples value.

### 2.15. Statistics

Continuous variables were expressed as median (interquartile range (IQR)) and comparisons between two groups were performed using a non-parametric Mann–Whitney U test, unless stated otherwise. Correlation analysis was carried out using a Spearman test. Analyses were performed with Statistica package (version13.5.0.17, TIBCO Software inc., Palo Alto, CA, USA). Statistical significance was defined at *p* < 0.05. For the in vitro results, statistical analysis was performed using the parametric unpaired Student’s *t*-test using GraphPad Prism (version 7.0., GraphPad SoftwareSan Diego, CA, USA). Statistical significance was defined as *p* < 0.05.

## 3. Results

### 3.1. Anti-ApoA-1 IgG Associations with Lipid Profile in FH Children

Baseline, clinical, and biological characteristics of FH and healthy children are described in Table 1. As expected, FH children had higher levels of total cholesterol, LDL cholesterol, apoB particles, HDL cholesterol, and miR-33a than their age-matched controls. No other differences in baseline clinical characteristics were observed between these two groups. Median anti-ApoA-1 IgG were similar in these two groups and a similar prevalence of high anti-ApoA-1 IgG levels was retrieved in both groups (38.8% and 37.9%, respectively).

In FH children, there were no significant differences between anti-ApoA-1 IgG positive and anti-ApoA-1 IgG negative children in terms of age, gender, and overweight-associated features, with the possible exception of BMI and waist circumference, which tended to be higher in ApoA-1 IgG positive FH children than seronegative ones. Anti-ApoA-1 IgG positive subjects showed significantly lower median levels of LDL cholesterol, total cholesterol, and miR-33a, with a marginal trend for apoB particles when compared to anti-ApoA1 IgG negative subjects (Table 2). Significant correlations were only observed between anti-ApoA-1 IgG and miR-33a levels (*r* = 0.42, *p* = 0.02).

In age-matched healthy children, the aforementioned differences between anti-ApoA-1 IgG seropositive and seronegative individuals related to lipid profile and miR-33a levels were not observed (Table 3). In contrasting to the observation in FH children, we observed that BMI, waist, and arm circumference in healthy children were lower in subjects positive for anti-ApoA-1 IgG when compared to seronegative children.

### 3.2. Lower Level of miR-33a in Anti-ApoA-1 IgG-Treated Human Macrophages

To further explore a possible causal link between the presence of anti-ApoA-1 antibodies, lower miR-33a levels, and a more favorable lipid profile in FH children, we first tested the ability of polyclonal anti-ApoA-1 IgG or control IgG to modulate the in vitro miR-33a production in human monocyte-derived macrophages (HMDM).

After a time course of 4, 8, 16, and 24 h of antibody exposure, miR-33a levels were detected by quantitative PCR, as shown in Figure 1a. Anti-ApoA-1 IgG, and not control IgG, was found to decrease the miR-33a production in vitro on HMDM. As shown in Figure 1a, when compared to baseline conditions and control IgG, anti-ApoA-1 IgG (40 µg/mL) induced a slight but consistent decrease of cellular miR-33a levels during each time point, with the exception of 4 h stimulation, and was statistically significant at 24 hours (*p* < 0.01). As shown in Figure 1b, this anti-ApoA-1 IgG-induced miR-33a downregulation was associated with a decreased miR-33a level at 24 h in the cell supernatant. Up to now, most of the anti-ApoA-1 IgG pro-atherogenic effects have been shown to be mediated by TLR2/TLR4/CD14 complex engagement followed by nuclear factor (NF)-kB and activator protein (AP)-1 pathway activation [4,14,17], and we examined the role TLR2/4, CD14, NF-kB, and AP-1 in the anti-ApoA-1 IgG-mediated downregulation of miR-33a. Accordingly, we pretreated these cells with blocking antibodies against TLR2, TLR4, and the co-receptor CD14, and added chemical inhibitors BAY11-7082 (5 µM) or SP600125 (20 µM), specific to NF-kB and AP-1, respectively, 30 minutes before treatment with polyclonal anti-ApoA-1 IgG for 24 h. As shown in Figure 1c, none of the used inhibitors modulated the anti-ApoA-1 IgG effect on miR-33a, suggesting that the anti-ApoA-1 IgG-dependent miR-33a regulation was independent of TLR2/4/CD14 complex signaling.

We then looked at the primary miR-33a (pri-miR-33a) precursor transcript that, after complex processing, gives the mature miR-33a, as well as at the SREBP2 mRNA, since miR-33a is located in the intronic region of *SREBP2* gene [23,37] and we checked whether they would follow the same regulation as for miR-33a. Surprisingly, HMDM exposure to anti-ApoA-1 IgG for 24 hours significantly increased pri-miR-33a levels as well as SREBP2 mRNA (Figure 2a,b).

These results suggest that despite the fact that the levels of primary miR-33a transcript are increased by anti-ApoA-1 IgG, the mature intracellular miR-33a form is nevertheless decreased by anti-ApoA-1 IgG exposure. Current ongoing investigations to dissect the exact mechanisms underlying this discrepancy indicate that miR-33a stability issues rather than maturation or nuclear export defects may be involved, but no clear-cut mechanisms have been identified so far.

### 3.3. Anti-ApoA-1 IgGs Modulate Cholesterol-Regulating Proteins in Macrophages

We further investigated the possible modulation, by anti-ApoA1 IgG, of SREBP2 transcription factor known to bind and activate *LDL-R* and *HMGCR* gene transcription [37,38,39] involved in cholesterol uptake and biosynthesis respectively. After 24 h of stimulation, anti-ApoA-1 IgG treatment increased the levels of SREBP2, HMGCR, and LDL-R protein when compared to the control IgG (Figure 3). As shown in Figure 3, only anti-ApoA-1 IgGs and not the control IgGs were found to increase the expression of the SCAP protein (SREBP cleavage-activating protein), responsible for cleaving and activating SREBP2 [40].

### 3.4. The Impact of Anti-ApoA-1 IgG on Cholesterol-Regulating Proteins Is Mediated by TLR2 and TLR4

Furthermore, we examined the role of TLR members as well as CD14, NF-kB, and AP-1 in the anti-ApoA-1 IgG-mediated upregulation of SREBP-2, LDL-R, and HMGCR on HMDM. Blocking both TLR2 and TLR4, NF-kB, and AP-1, as previously described, induced a significant reduction in the SREBP-2, LDL-R, and HMGCR protein expression (Figure 4b–d), while only blocking AP-1 significantly decreased anti-ApoA-1 IgG-induced HMGCR protein expression (Figure 4d).

### 3.5. Anti-ApoA-1 IgGs Promote LDL Uptake and Foam Cell Formation

The complexity underlined by the previous observations indicating that i) anti-ApoA-1 IgG are associated with a more favorable lipid profile in FH children but lower miR-33a levels and ii) that these antibodies could increase both pro-atherogenic (HMGCR increase) and anti-atherogenic pathways (SREBP-2, LDLR) in vitro prompted us to evaluate the global result of exposing human macrophages to anti-ApoA-1 IgG in terms of foam cell formation. For this purpose, HMDM were treated with or without LDL (20 µg/mL) in the presence or absence of anti-ApoA-1 IgG (40 µg/mL) or control IgG (40 µg/mL) for 24 hours and followed by Oil Red O staining of HMDM. As shown in Figure 5, anti-ApoA-1 IgG exposure without LDL induced a modest but significant increase in cellular lipid content when compared to control IgG (Figure 5b). Furthermore, in the presence of LDL, the effect of anti-ApoA-1 IgG was enhanced (Figure 5a,b). In presence of LDL, the anti-ApoA-1 IgG effect on foam cell formation was abrogated by all the inhibitors used, with the exception of AP-1 inhibitor, which nevertheless provided a close to significant inhibition (Figure 5b). Taken together, these results indicate that anti-ApoA-1 IgG promotes foam cell formation through partly TLR2/4-dependent pathways. Of note, this anti-ApoA-1 IgG-induced foam cell formation was specifically enhanced by native LDL because in presence of oxidized LDL, anti-ApoA-1 IgG increases foam cell formation to the same extent as anti-ApoA-1 IgG alone (Appendix A).

Importantly, given the fact that LDL per se substantially induced foam cell formation regardless of the concomitant presence of anti-ApoA-1 IgG, the rest of our experimental procedures were carried out without LDL in order to assess the effects specifically ascribed to anti-ApoA-1 IgGs.

### 3.6. Anti-ApoA-1 IgGs Upregulate ABCA1

Because ABCA1 is a known target of miR-33a [23], we expected that the anti-ApoA-1 IgG-induced downregulation of miR-33a would increase ABCA1 expression. As shown in Figure 6, anti-ApoA-1 IgG increased both the mRNA (Figure 6a) and protein levels of ABCA1 (Figure 6b) in HMDM.

### 3.7. ACAT Activity and Cellular Cholesterol Distribution in Macrophages Following Anti-ApoA1 IgG Treatment

Because ABCA1 levels are known to be upregulated as part of a homeostatic feedback loop in presence of intracellular lipid overload [36], the previous experiments did not allow us to conclude whether there was a direct or indirect effect of anti-ApoA-1 IgG on ABCA1 levels. Nevertheless, as the net effect of anti-ApoA-1 IgGs was an increase in foam cell formation (which is usually associated with lower ABCA1 levels), we primarily suspected that the ABCA1 increase in response to anti-ApoA-1 IgG was part of a negative feedback mechanism. In that case, the exposure of anti-ApoA-1 IgG on HMDM should primarily reorient the free cholesterol present in the plasmatic membrane toward esterified cholesterol intracellular pools, which is mediated by the acyl coenzyme A: cholesterol acyltransferase (ACAT) [41]. To validate this hypothesis, we assessed the impact of anti-ApoA-1 IgGs on the ACAT activity and protein levels on HMDM. As shown in Figure 7a, anti-ApoA-1 IgGs increased ACAT activity, while the protein levels remained unchanged (Appendix A).

As shown in Figure 7b, this anti-ApoA-1 IgG- induced ACAT activation was accompanied by a significant reduction of free cholesterol content in membrane and a non-significant reduction of the free cholesterol content in the cell supernatant counterpart (Figure 7c).

### 3.8. Anti-ApoA-1 IgG Impact on Cholesterol Passive Diffusion

Cholesterol passive diffusion (also known as aqueous diffusion) represents the free cholesterol exchange occurring between cells membranes and plasma by a passive process driven by free cholesterol concentration gradients towards HDL/ApoA-1, independently of any efflux pumps [36] In normocholesterolemic conditions, passive diffusion has been shown to account for up to 80% total cholesterol macrophage efflux [42]. Given the decrease in membrane free cholesterol induced by anti-ApoA-1 IgG in response to the ACAT redistribution within intracellular pools, we expected these antibodies to decrease passive diffusion due to the decrease in the free cholesterol gradient. For this purpose, we used a validated model of human macrophage derived from THP-1 monocytic cell line [35] as our human primary macrophage model was not validated for such kind of experiments. Cells were treated for 24 h with anti-ApoA 1 IgGs or control IgGs in the absence of serum but in presence of 0.2% albumin to avoid any active ABCA1-, ABCG1- or SR-B1-mediated cholesterol efflux. As shown in Figure 8, anti-ApoA-1 IgGs significantly decreased the passive diffusion.

## 4. Discussion

In this study, we report, for the first time, that the prevalence of high anti-ApoA-1 IgG levels in children devoid of any concomitant autoimmune diseases is substantial, reaching 38%, and seemed unrelated to familial hypercholesterolemia. Such a prevalence is similar to what has been reported in adults in secondary prevention or equivalent CV risk-associated conditions [4,7,43], and close to double of what has been observed in a general adult population [3], where these antibodies were shown to be associated with a worse overall and CV prognosis [3,5].

The second notable finding of this study is that FH positive children with high anti-ApoA-1 IgG levels displayed a more favorable lipid profile consisting of lower total and LDL cholesterol levels, when compared to children who tested negative for these autoantibodies. These results complement previous observations [3,13,15,16] showing that high anti-ApoA-1 IgG levels were associated with lower total, LDL, and miR-33a levels, despite being associated with increased CVD risk, according to previous reports [3,13,15,16]. This clinical observation derived from a limited number of pediatric FH patients (and not retrieved on age and gender matched controls), prompted us to evaluate the ability of anti-ApoA-1 IgG to promote in vitro foam cell development on human primary macrophages, and to determine the molecular mechanisms involved, which led to the following new mechanistic findings.

If the role of the TLR2/4/CD14 complex has been well documented accounting for the cytokine-dependent pro-inflammatory, pro-thrombotic, and pro-arrythmogenic biological properties ascribed to anti-ApoA-1 IgG [9,14,17], this is the first demonstration that these antibodies promote foam cell formation on human macrophages by the same innate immune receptors. Our results demonstrate that anti-ApoA-1 IgGs upregulate the expression of key cholesterol homeostasis-related proteins, such as SREBP2, LDL-R, and HMGCR in a TLR2/4 complex-dependent manner, leading to both increased LDL uptake and intracellular cholesterol synthesis, independently of oxidized LDL uptake. We considered this cellular cholesterol overload as the key driver of ACAT stimulation [44] leading to the esterification of free cholesterol excess into intracellular cholesterol pools (lipid droplet accumulation) and decreased the amount of plasma membrane free cholesterol, diminishing the free cholesterol gradient requested for passive diffusion, a key macrophage cholesterol efflux regulator in normolipidemic conditions [36]. Our observations are in line with previous observations demonstrating that TLR4 and TLR2 agonists can upregulate SREBP2, HMGCR, and LDL-R expression and promote foam cell formation independently of oxidized-LDL uptake [29,45,46].

In this context, the anti-ApoA-1 IgG-induced increase of the anti-atherogenic ABCA1 protein expression, and decrease in pro-atherogenic miR-33a has to be understood as the reflection of the activation of a compensatory but insufficient homeostatic feedback loop to limit intracellular cholesterol overload for two main reasons. First of all, our results indicate that miR-33a expression is independent of TLR2/TLR4 signaling, whereas TLR2 and 4 ligands have been reported to downregulate miR-33a in the case of direct signaling [47]. Secondly, any increase in intracellular cholesterol accumulation is known to downregulate miR-33a expression which, in turn, will alleviate miR-33a-mediated ABCA1 inhibition, leading to an increase in ABCA1 protein expression and ABCA1 efflux independently of TLR stimulation [23,48]. Taken together, these results indicate that in our experimental context, changes in miR-33a and ABCA1 expression have to be considered as the effectors of a negative feedback loop aimed at limiting anti-ApoA-1 IgG-induced intracellular lipid accumulation without being able to fully prevent it, as a common feature of negative feedback loops [49]. Nevertheless, our results indicate that in presence of anti-ApoA-1 IgG, the usual negative feedback mechanism activated by high intracellular cholesterol levels to reduce LDLR seems to be blunted [49]. The exact mechanisms underlying this observation are currently unknown and warrant further investigations.

Despite these remaining questions, to the best of our knowledge, this is the first thorough experimental demonstration unravelling the detailed mechanisms by which anti-ApoA-1 IgGs could promote atherogenesis by acting as disruptors of macrophage cholesterol homeostasis by increasing both LDL uptake and intracellular cholesterol synthesis independently of oxidized LDL. As such, our results show that anti-ApoA-1 IgGs have at least two antagonist mechanisms of action to those of statins in the sense that the former act as HMGCR and ACAT activators, whereas statins act as HMGCR and ACAT inhibitors [50] Although, concordant with our clinical data, knowing whether these mechanisms could explain the clinical associations retrieved between anti-ApoA-1 IgG and lipid profile remains to be determined. Our current understanding of these possible pathophysiological mechanism is summarized in Appendix A.

This study has several limitations. It is worth mentioning the very limited size of our FH pediatric sample used in this translational study, due to the low prevalence of FH [51] and the issues related to pediatric blood sampling. Despite this power limitation, the significant associations derived from this cohort were instrumental in providing guidance for the in vitro experiments which, in turn, provided results that could explain the reported clinical associations, albeit not providing a formal demonstration. Another limitation resides in the unresolved paradox raised by the fact that anti-ApoA-1 IgG can decrease mature miR-33a levels despite increasing levels of miR-33a precursor. The reason underlying such an observation is under active investigation, as we currently suspect a possible effect of anti-ApoA-1 IgG on miRNA decay that we did not investigate due to highly elusive mechanisms underlying miRNA stability [52,53], which could not be resolved within the scope of the present work. The third limitation is that we did not consider ABCA1 efflux in this work because all our experiments related to protein regulation were done in normocholesterolemic conditions without additional LDL, where passive diffusion is believed the main functional driver of the cholesterol efflux in macrophages [36]. Nevertheless, as ABCA1 levels were increased in response to anti-ApoA-1 IgG as part of the negative feedback loop (see above), we expect ABCA1 cholesterol efflux to be concomitantly increased, but not sufficiently to inhibit anti-ApoA-1 IgG-induced foam cell formation. Lastly, due to limited human sample availability, we could not evaluate the effect of human purified anti-ApoA-1 IgG in this study. Nevertheless, as we previously demonstrated that the commercial anti-human ApoA-1 IgG used in this study promoted the same effects as the human-purified IgG fraction containing high levels of these autoantibodies in vitro [9,17], and because these autoantibodies have been consistently used in different validated and published animal and in vitro studies [9,12,13,14,17], we expect a similar impact on in vitro lipid metabolism to occur by using human purified anti-ApoA-1 IgG, even if not formally demonstrated.

In conclusion, the results of this hypothesis-generating translational study show that the prevalence of ApoA-1 IgG seropositivity is frequent in children, unrelated to FH, and surprisingly associated with a favorable lipid profile in FH but not in controls. Furthermore, anti-ApoA-1 IgG were found to induce foam cell formation in vitro through a complex interplay between innate immune receptors and key cholesterol homeostasis regulators, functionally impairing the cholesterol efflux capacity of macrophages. The clinical implications of these findings are unclear, but may suggest that the presence of circulating anti-ApoA-1 IgG could not only interfere with some beneficial effects of lipid-lowering drugs in humans, but could also potentially decrease the contribution of the lipid profile to the individual’s CV risk in current CV risk stratification tools. If so, these antibodies may be considered as an additional CV risk enhancer to the list of the factors already considered as such in the latest dyslipidemia management guidelines [54]. These hypotheses are currently devoid of any experimental evidence and await further investigations.

## Figures and Tables

**Figure 1 jcm-08-02035-f001:**
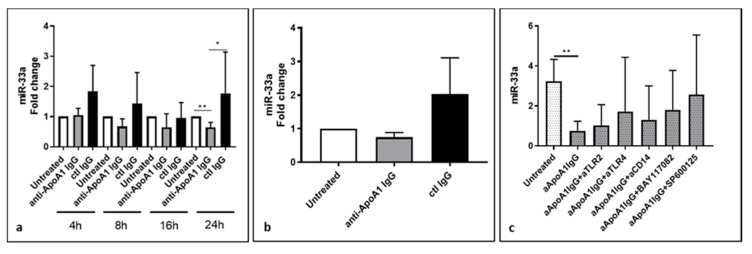
Lower levels of intracellular and extracellular miR-33a in anti-ApoA-1 IgG-treated human macrophage. Cells were treated with anti-ApoA-1 IgG or control antibodies for 4, 8, 16, and 24 h and RT-PCR was performed to determine (**a**) intracellular miR-33a levels as well as (**b**) miR-33a in the supernatant counterpart after 24 h anti-ApoA-1 IgG or control IgG treatment. (**c**) Blocking antibodies againstTLR2/4 were used as well as inhibitors of NF-kB and AP-1 to investigate the anti-ApoA-1 IgG mediated miR-33a downregulation. In (**a**,**b**), data are expressed as fold change expression of the mean ± SD of miR-33a calculated by ΔΔCT method of five independent experiments (*n* = 5) and values were normalized to untreated condition, while in panel (**c**), data are expressed as miR-33a quantity (2^−ΔΔCt^). *p*-values were calculated using Student’s *t*-test. Panel a, * *p* = 0.015, ** *p* = 0.0012. Panel b, the difference is not statistically significant, anti-ApoA-1 IgG vs. untreated *p* = 0.11, anti-ApoA-1 IgG vs. ctl control IgG *p* = 0.28. (**c**), ** *p* = 0.0017.

**Figure 2 jcm-08-02035-f002:**
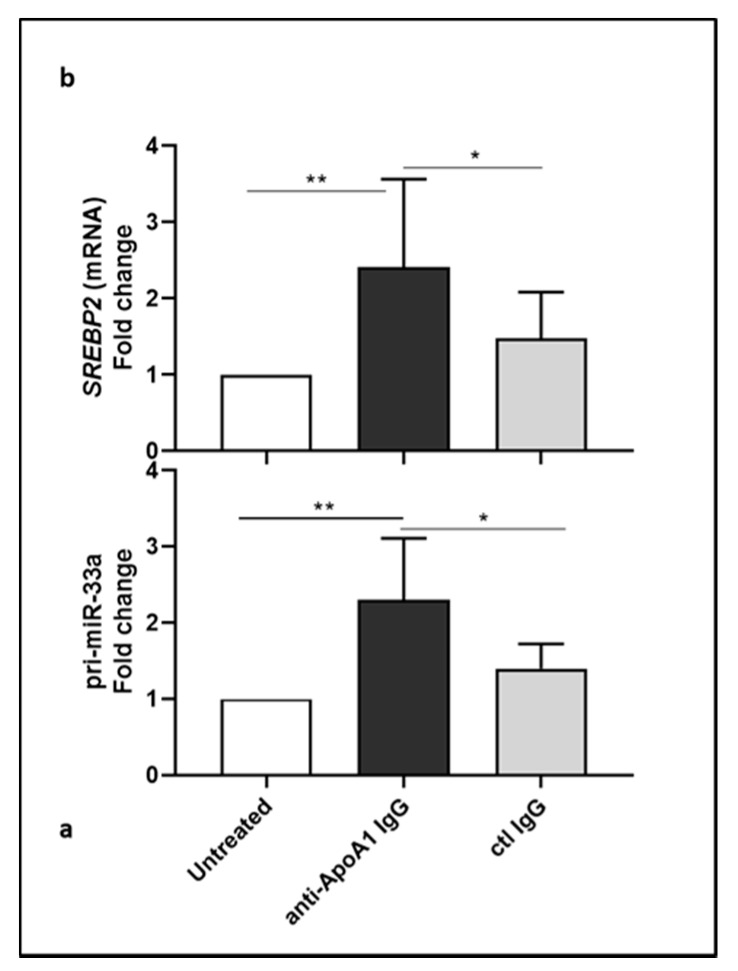
Primary miR-33a and SREBP2 mRNA are upregulated by anti-ApoA-1 IgG. Cells were treated with anti-ApoA-1 IgG or control antibodies for 24 h and RT-PCR was performed to determine pri-miR-33a and SREBP2 mRNA levels (**a**,**b**). Data are expressed as fold change expression of the mean ± SD of pri-miR-33a or SREBP2 mRNA calculated by ΔΔCT method of five independent experiments (*n* = 5) and values were normalized to untreated condition. *p*-values were calculated using the Student’s *t*-test. (**a**), * *p* = 0.032, ** *p* = 0.0031; (**b**), * *p* = 0.043, ** *p* = 0.007.

**Figure 3 jcm-08-02035-f003:**
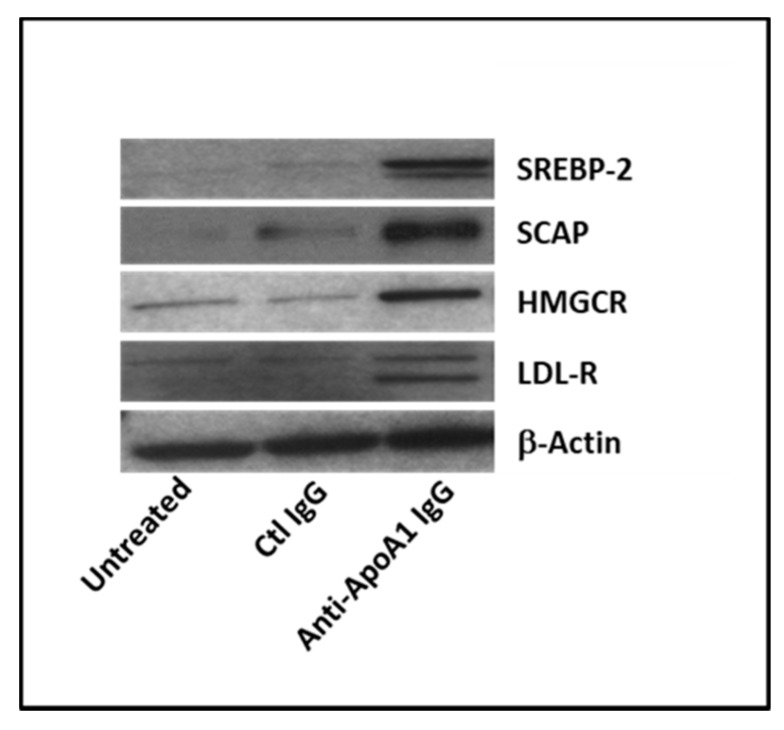
Anti-apoA-1 IgGs interfere with key regulators of cholesterol metabolism in macrophages. Anti-ApoA-1 IgGs induced the expression of the key regulators of the cholesterol pathway after 24 h as evidenced by Western blot. One of four representative Western blot is shown (*n* = 4).

**Figure 4 jcm-08-02035-f004:**
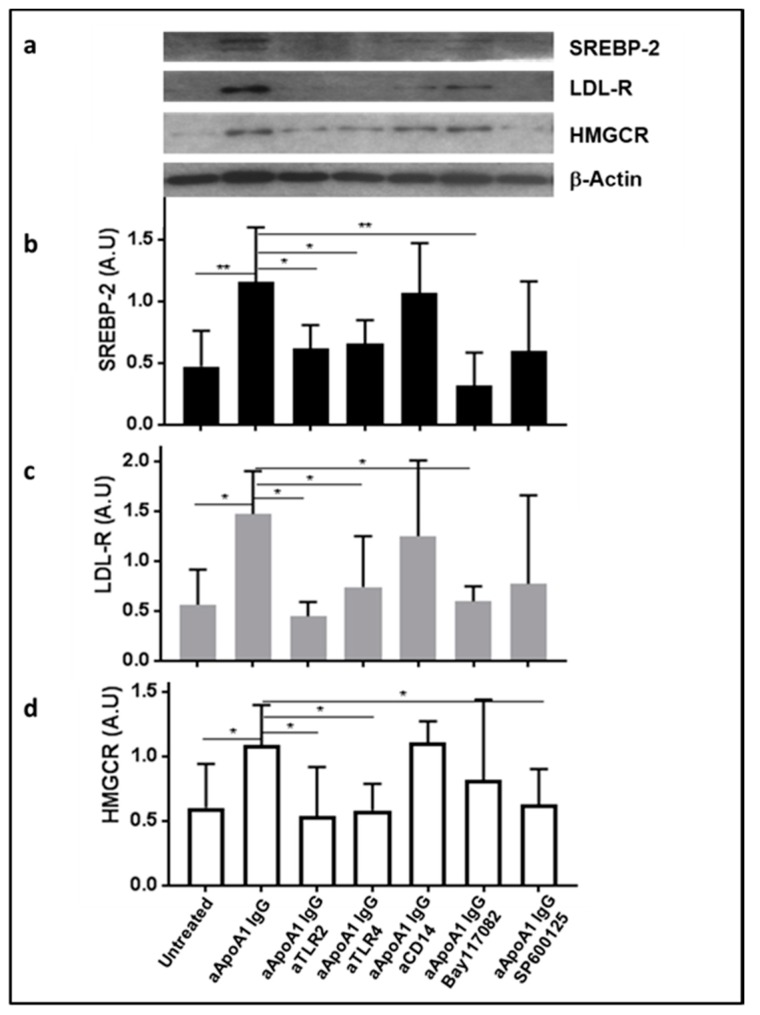
The impact of anti-ApoA-1 IgG on cholesterol metabolism is mediated by TLR2, TLR4, Nf-KB, and AP-1 transcription factors. Western blot-derived results showed that anti-ApoA-1 IgG-mediated upregulation of SREBP2, LDL-R, and HMGCR on human monocyte-derived macrophages (HMDM) is significantly reduced by blocking TLR2 and TLR4 with specific blocking antibodies. Blocking NF-kB and AP-1 using the specific inhibitors BAY117082 and SP600125 respectively provided similar results. (**a**) One of five representative Western blots is shown. (**b**–**d**) Data are expressed as mean ± SD of band intensity volume/actin intensity volume from five different experiments (*n* = 5). *p*-values were calculated using the Student’s *t*-test: * *p* < 0.05, ** *p* < 0.01.

**Figure 5 jcm-08-02035-f005:**
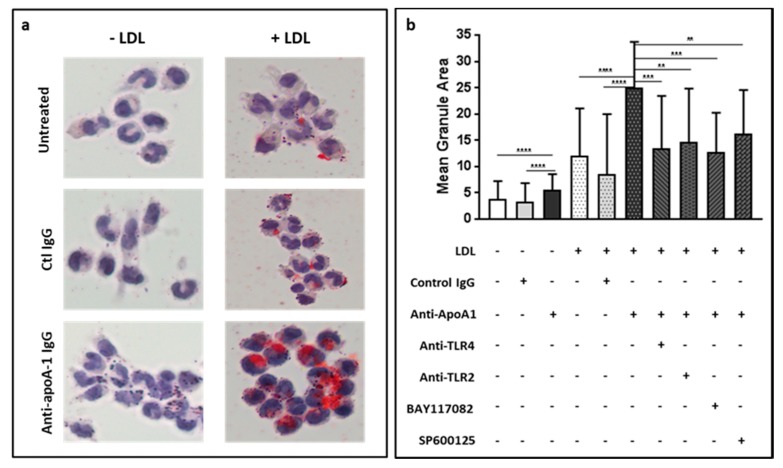
Anti-ApoA-1 IgG promote LDL uptake and foam cell formation mediated by TLR2/4, NF-kB, and AP1. (**a**) HMDM were treated for 24 h with anti-ApoA-1 IgG or ctl IgG in the presence or absence of native LDL (20 µg/mL). Cells were stained with Oil Red O to highlight the lipid uptake. (**b**) Blocking antibodies to TLR2/4 were used as well as inhibitors to NF-kB and AP-1 to try to inhibit the anti-ApoA-1 IgG mediated LDL uptake as evidenced by Oil Red O staining quantification as the mean granule area per cell. Oil Red O was quantified as indicated in the method section. Results are expressed in arbitrary units as mean ± SD of four independent experiments (*n* = 4), *p*-values were calculated using the Student’s *t*-test: *** p* < 0.01, **** p* < 0.001, **** *p* < 0.0001.

**Figure 6 jcm-08-02035-f006:**
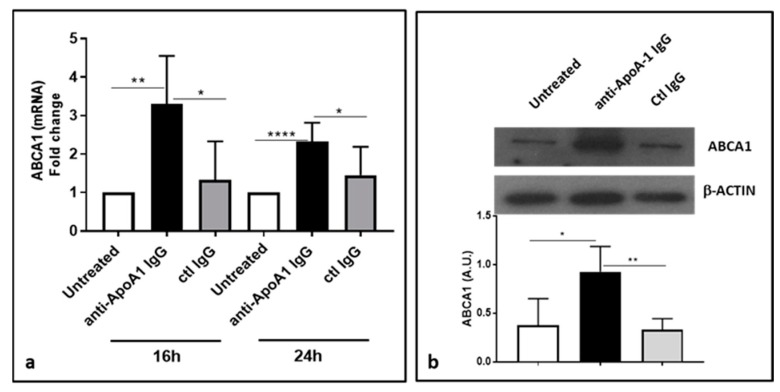
Anti-ApoA-1 IgGs upregulate ABCA1. ABCA1 was increased after 16 and 24 h of anti-ApoA-1 IgG exposure to HMDM at the mRNA level as revealed by RT-PCR (**a**) as well as at the protein level after 24 h anti-apoA-1 IgG stimulation, as revealed by Western blot analysis (**b**). In panel a, data are expressed as fold change expression of the mean ± SD of ABCA1 calculated by ΔΔCT method of six independent experiments (*n* = 6) and values were normalized to untreated condition. *p*-values were calculated using the Student’s *t*-test: * *p* < 0.05, ** *p* < 0.01, **** *p* < 0.0001. (**b**) One of four representative Western blots is shown. Data are the mean ± SD of band intensity volume/actin intensity volume of four independent experiments (*n* = 4). *p*-values were calculated using the Student’s *t*-test: * *p* = 0.02, ** *p* = 0.005.

**Figure 7 jcm-08-02035-f007:**
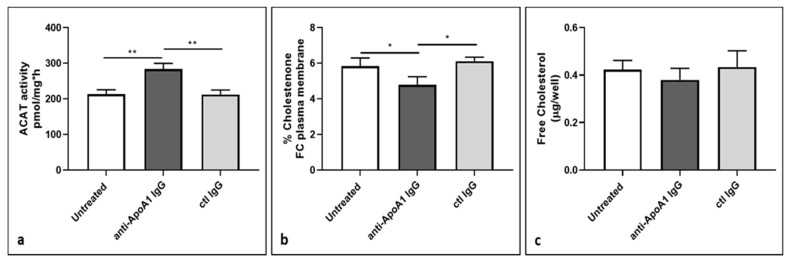
Effect of anti-ApoA-1 IgG on cholesterol distribution. (**a**) Cells were incubated for 24 hours in RPMI medium with 10% FCS, control IgGs, or anti-ApoA-1 IgG. Monolayers underwent a second incubation (4 h) in the presence of [1-14C]-oleic acid albumin complex. Data are expressed as the mean ± SD of the measurements done in triplicate and repeated on three HMDM donors (*n* = 3). Statistical differences were determined by Student’s *t*-test: ** *p* ≤ 0.0036. (**b**) HMDM were labelled with 3 μCi/mL [3H] cholesterol for 24 h in RPMI medium with 10% FCS, control IgGs or anti-ApoA-1 IgGs. Cells were then washed and incubated with 1 U/mL cholesterol oxidase enzyme in DPBS for 4 hours at 37 °C. Data are expressed as the mean ± SD of the measurements done in triplicate and repeated on three HMDM donors (*n* = 3), *p*-values were calculated using the Student’s *t*-test: * *p* < 0.05. (**c**) Free cholesterol content in HMDM supernatant after exposure of cells to anti-ApoA-1 IgGs or control IgGs was expressed as µg of free cholesterol per well. Data are expressed as the mean ± SD of the measurements done on five HMDM donors (*n* = 6). The differences between groups were not statistically significant, anti-ApoA-1 IgG vs. untreated *p* = 0.12; anti-ApoA-1 IgG vs. ctl IgG *p* = 0.14.

**Figure 8 jcm-08-02035-f008:**
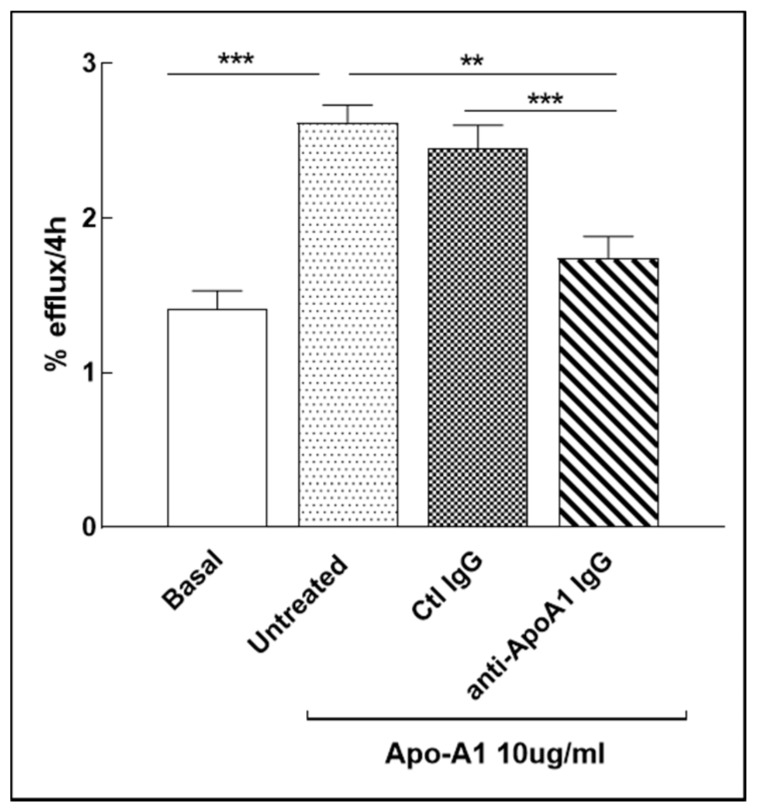
Anti-ApoA-1 IgGs inhibit passive diffusion. THP-1 cells were treated for 24 h with anti-ApoA-1 IgGs or ctl IgGs and passive diffusion was analyzed and expressed as percentage of the radioactivity released to the medium in 4 h over the total radioactivity incorporated by cells. *p*-values were calculated using the Student’s *t*-test: ** *p* = 0.010, *** *p* < 0.001.

**Table 1 jcm-08-02035-t001:** Baseline demographic and biological characteristics of subjects according to familial hypercholesterolemia (FH) status.

Characteristic	Healthy Children (*n* = 25)	FH Children (*n* = 29)	* *p*-Value
Age, year (IQR)	8.5 (7–11)	9 (6–11)	0.89
Sex (male %)	45	48.2	0.88
BMI (kg/m^2^)	19 (16.6–20.2)	17.8 (15.6–20)	0.55
Systolic BP *, mmHg (IQR)	105 (95–110)	110 (100–120)	0.14
Diastolic BP, mmHg (IQR)	60 (60–70)	60 (60–70)	0.75
Waist circumference, cm (IQR)	61.5 (53–69)	60 (53–68)	0.94
Hip circumference, cm (IQR)	66 (61–76)	64 (57–73)	0.85
Arm circumference, cm (IQR)	20 (18–22)	20 (17–23)	0.83
Total cholesterol (mg/dL)	152.5 (138–167.5)	231 (213–280)	<0.0001
LDL cholesterol (mg/dL)	91.4 (79.5–103)	156 (137.6–211.6)	<0.0001
HDL cholesterol (mg/dL)	52 (43.5–58.5)	57 (51–66.5)	0.02
Triglycerides (mg/dL)	56.5 (45.5–63)	64 (47–75)	0.17
Apolipoprotein B (mg/dL)	65 (49–81.5)	93 (86–134)	<0.0001
Anti-ApoA-1 IgG (OD value)	0.53 (0.4–0.77)	0.55 (0.5–0.68)	1
Anti-ApoA-1 IgG positivity (%)	38.8	37.9	0.9
miR-33a (2^-ΔCt)	0.05 (0.03–0.2)	0.4 (0.19–1)	<0.0001

* *p*-values were obtained by comparing FH positive versus FH negative subjects, *p*-values were calculated using the Mann–Whitney U test. BMI, body mass index. BP, blood pressure. IQR: interquartile range.

**Table 2 jcm-08-02035-t002:** Baseline demographic and biological characteristics of FH children according to anti-ApoA-1 IgG status.

Characteristic	Healthy Children (*n* = 25)	Anti-ApoA-1 Negative (*n* = 15)	Anti-ApoA-1 Positive (*n* = 10)	* *p*-Value
Age, year (IQR)	8.5 (7–11)	10 (7–11)	7 (7–9)	0.08
Sex (male %)	45	42.8	50	0.28
BMI (kg/m^2^)	19 (16.6–20.2)	19.2 (17–20.2)	17.2 (15.4–19.5)	0.02
Systolic BP *, mmHg (IQR)	105 (95–110)	100 (90–105)	110 (100–120)	0.7
Diastolic BP, mmHg (IQR)	60 (60–70)	60 (60–75)	62.5 (60–70)	0.53
Waist circumference, cm (IQR)	61.5 (53–69)	66 (51–69)	61 (54–64)	0.03
Hip circumference, cm (IQR)	66 (61–76)	70 (61–76)	66 (61–67)	0.09
Arm circumference, cm (IQR)	20 (18–22)	20 (17–22)	19.5 (18–21)	0.01
Total cholesterol (mg/dL)	151 (138.5–167.5)	147 (135–167)	163 (148–168)	0.27
LDL cholesterol (mg/dL)	91.4 (79.5–103)	89.4 (72–95)	101.8 (87–108)	0.11
HDL cholesterol (mg/dL)	52 (43.5–58.5)	50.5 (39–55)	56 (47–62)	0.12
Triglycerides (mg/dL)	56.5 (45.5–63)	58.5 (47–70)	55 (41–58)	0.97
Apolipoprotein B (mg/dL)	65 (49–81.5)	53.5 (40–66)	79.5 (69–85)	0.19
miR-33a (2^-ΔCt)	0.05 (0.03–0.25)	0.1 (0.03–0.26)	0.05 (0.04–0.1)	0.4

* *p*-values were obtained by comparing anti-ApoA-1 IgG positive versus anti-ApoA-1 IgG negative subjects, *p*-values were calculated using the Mann–Whitney U test. IQR: interquartile range.

**Table 3 jcm-08-02035-t003:** Baseline demographic and biological characteristics of healthy children according to anti-ApoA-1 IgG status.

Characteristic	FH Children (*n* = 29)	Anti-ApoA-1 Negative (*n* = 18)	Anti-ApoA-1 Positive (*n* = 11)	* *p*-Value
Age, year (IQR)	9 (6–11)	8 (7–10)	10 (6–13)	0.45
Sex (male %)	48	54.5	72.7	0.08
BMI (kg/m^2^)	17.8 (15.6–20)	16.8 (15.3–18.2)	19.3 (17.4–21.9)	0.05
Systolic BP^*^, mmHg (IQR)	110 (100–120)	110 (100–115)	110 (100–125)	0.39
Diastolic BP, mmHg (IQR)	60 (60–70)	62.5 (60–70)	60 (60–70)	0.58
Waist circumference, cm (IQR)	60 (53–68)	56.5 (53–63)	66 (61–70)	0.05
Hip circumference, cm (IQR)	64 (57–73)	72 (64–80)	62.5 (57–71)	0.10
Arm circumference, cm (IQR)	20 (17–23)	22 (22–23)	19 (17–22)	0.14
Total cholesterol (mg/dL)	213 (213–280)	263 (219–305)	209 (188–240)	0.04
LDL cholesterol (mg/dL)	156.6 (137.6–211.6)	191.2 (144.6–228)	124.4 (113.8–168.8)	0.04
HDL cholesterol (mg/dL)	57 (51–66.5)	61 (55–67)	55 (50–58)	0.24
Triglycerides mg/dL)	64 (47–75)	60 (47–66)	66 (35–78)	0.35
Apolipoprotein B (mg/dL)	93 (86–134)	109 (91–143)	86 (76–103)	0.05
miR-33a (2^-Ct)	0.41(0.19–1.03)	0.85 (0.36–1.2)	0.28 (0.14–0.5)	0.04

* *p*-values were obtained by comparing anti-ApoA-1 IgG positive versus anti-ApoA-1 IgG negative subjects, *p*-values were calculated using the Mann–Whitney U test. IQR: interquartile range.

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
