# Peer review of "Anti-ApoA-1 IgGs in Familial Hypercholesterolemia Display Paradoxical Associations with Lipid Profile and Promote Foam Cell Formation"

_jcm, 2019, doi:10.3390/jcm8122035_

Round 1

Reviewer 1 Report

Considering the changes that the authors made on the manuscript and the explanations given to my concerns, the paper is now ready for publication.

Reviewer 2 Report

Authors have answered all my comments

I do not have further comments.

This manuscript is a resubmission of an earlier submission. The following is a list of the peer review reports and author responses from that submission.

Round 1

Reviewer 1 Report

The article explores the association of the presence in blood of Apo A-1 antibodies in a group of children with familial hypercholesterolemia and in a normolipemic control group. As previously described in the same group, the presence of anti-apo A-1 antibodies is associated with a different lipid phenotype, especially a reduction in LDL cholesterol.

The second component of the study is to identify the mechanisms of this association. The conclusions are that antibodies induce the formation of foam-cells via toll-like receptors.

The clinical part is of little relevance. The number of subjects is small, the results are not significant, the results do not show a clear pattern of association, which is different between children with FH and normolipemics.

The definition of positivity of antibodies is arbitrary, and a continuous variable is transformed into dichotomous.

The mechanisms of the appearance of these antibodies in the blood are not explored, nor are they discussed.

The second part, however, is very robust with in vitro experiments that give novel information about the mechanisms by which antibodies are associated with increased risk of arteriosclerosis, and associate the formation of foamy cells.

The only objection to the article is that the second part of the work, which is the most important, seems to fit little to the scope of the journal.

Reviewer 2 Report

The work entitled "Antibodies against Apolipoproteina A-1 as disruptors of cellular cholesterol homeostasis promoting foam cell formation" exlpores how anti-ApoA-1 IgG disrupt cholesterol homeostasis.

The in vitro studies seem appropiate and the results interesting, however, the work in its current form present a huge drawback as I don't see how studying pediatric patients with or without  familial hypercholesterolemia relates with the in vitro data. It seems like two independent works to me. I do not see how the associations derived from studying the FH cohort provide guidence for in vitro experiments as stated in the discussion section. Moreover, if including pediatric patients with FH in the study is important, it should be addressed in the title or in the introduction. How would you justify the inclusion of this population? How can pediatric patients with FH with no clinical sympomatology (inclusion criteria only based on total cholesterol levels and relatives with FH according to MEDPED punctuation) help in elucidating the role of anti-ApoA1 IgG in disrupting cholesterol homeostasis?Furthermore, including the cohort does not help on your results since Anti-ApoA-1 IgG positive patients present with a favorable lipid profile, which contradicts your in vitro results.   

I would suggest foccusing on in vitro data for publication and forgetting about the pediatric human data.               

Also, the limitations included in the discussion section are too numerous and deserve further elucidation.